# Could the Human Endogenous Retrovirus-Derived Syncytialization Inhibitor, Suppressyn, Limit Heterotypic Cell Fusion Events in the Decidua?

**DOI:** 10.3390/ijms221910259

**Published:** 2021-09-23

**Authors:** Jun Sugimoto, Sehee Choi, Megan A. Sheridan, Iemasa Koh, Yoshiki Kudo, Danny J. Schust

**Affiliations:** 1Department of Obstetrics and Gynecology, Graduate School of Biomedical & Health Science, Hiroshima University, 1-2-3 Kasumi, Minami-ku, Hiroshima 734-8551, Japan; juns@hiroshima-u.ac.jp (J.S.); iemasako@hiroshima-u.ac.jp (I.K.); yoshkudo@hiroshima-u.ac.jp (Y.K.); 2Department of Obstetrics, Gynecology and Women’s Health, Division of Reproductive Endocrinology and Infertility, University of Missouri, 500 North Keene Street, Suite 203, Columbia, MO 65201, USA; seheechoi@missouri.edu (S.C.); mahrmf@health.missouri.edu (M.A.S.)

**Keywords:** suppressyn, human endogenous retrovirus, placenta, fusion, decidua

## Abstract

Proper placental development relies on tightly regulated trophoblast differentiation and interaction with maternal cells. Human endogenous retroviruses (HERVs) play an integral role in modulating cell fusion events in the trophoblast cells of the developing placenta. Syncytin-1 (ERVW-1) and its receptor, solute-linked carrier family A member 5 (SLC1A5/ASCT2), promote fusion of cytotrophoblast (CTB) cells to generate the multi-nucleated syncytiotrophoblast (STB) layer which is in direct contact with maternal blood. Another HERV-derived protein known as Suppressyn (ERVH48-1/SUPYN) is implicated in anti-fusogenic events as it shares the common receptor with ERVW-1. Here, we explore primary tissue and publicly available datasets to determine the distribution of ERVW-1, ERVH48-1 and SLC1A5 expression at the maternal-fetal interface. While SLC1A5 is broadly expressed in placental and decidual cell types, ERVW-1 and ERVH48-1 are confined to trophoblast cell types. ERVH48-1 displays higher expression levels in CTB and extravillous trophoblast, than in STB, while ERVW-1 is generally highest in STB. We have demonstrated through gene targeting studies that suppressyn has the ability to prevent ERVW-1-induced fusion events in co-culture models of trophoblast cell/maternal endometrial cell interactions. These findings suggest that differential HERV expression is vital to control fusion and anti-fusogenic events in the placenta and consequently, any imbalance or dysregulation in HERV expression may contribute to adverse pregnancy outcomes.

## 1. Introduction

Abnormalities in human placental development have been closely linked to pregnancy loss, pre-eclampsia/eclampsia, fetal growth retardation and placenta accreta and percreta [1,2,3,4,5]. Proper placental development involves a delicate balance between fusion of cytotrophoblast cells (CTB) into the syncytiotrophoblast (STB) layer and apoptotic shedding of syncytiotrophoblast material into the maternal circulation and tight regulation of trophoblast invasion into the maternal decidua/myometrium [3]. In the human placenta, trophoblast syncytialization is regulated by proteins derived from human endogenous retroviruses (HERVs) [6,7,8,9,10]. Two HERV proteins, syncytin-1 (syn1; ERVW-1) and syncytin-2 (syn2; ERVFRD1) and their receptors, solute-linked carrier family A member 5 (SLC1A5, aka ASCT2) and major facilitator superfamily domain-containing protein 2 (MFSD2), respectively, have been specifically implicated in fusion events [11,12,13,14]. We have recently described a third placental HERV protein, called suppressyn (SUPYN), that inhibits the profusogenic activity of syn1, but not syn2, in a dose-dependent fashion in in vitro models of CTB fusion [15,16].

Suppressyn is a truncated envelope protein derived from the HERV-H family of viruses (*ERVH48-1*) that is expressed at multiple sites in the placenta, including the CTB, and to a lesser extent the STB, of the floating villi, in the intermediate CTB of the anchoring villi, in invasive EVT (extravillous cytotrophoblast cells) within the decidua and in the endovascular trophoblast (endoTB) lining the maternal decidual vessels [15,16,17]. Since both cell-associated and secreted forms of suppressyn bind to SLC1A5, the cell surface receptor for syn1 [15], we have hypothesized that the anti-fusogenic effects of suppressyn on syn1-induced trophoblast fusion occur at the level of this shared binding partner. For intracellular suppressyn, this appears to involve suppressyn-associated alterations in SLC1A5 glycosylation, likely during intracellular processing of SLC1A5 for cell surface expression [15,16]. This, in turn, may alter the physical properties of the receptor and/or its expression levels or stability at the cell surface. Secreted suppressyn most likely inhibits syn1-mediated TB fusion via direct receptor interference, a property common to retroviruses that share a surface entry receptor [15]. 

We have shown that, similar to suppressyn itself, the syn1/suppressyn receptor, SLC1A5, is detected mainly in the CTB of the placental chorionic villi where CTB and STB fusion occurs. It is these interactions between trophoblast subtypes that are modeled by our prior mechanistic studies [15,16,17]. Localization of suppressyn at other sites within the maternal fetal interface [16], however, suggested a potential role in modulating fusion between trophoblast and non-trophoblast cells at the maternal-fetal interface. These might include fusion of EVT with EVT, of EVT with decidual glandular epithelial, stromal, or interstitial immune cells or possibly of endoTB with maternal endothelial or perivascular smooth muscle cells. Here, we aimed to confirm prior reports [18] on decidual protein expression of SLC1A5 and to use publicly available trascriptomic data to assess patterns of expression of trophoblast *ERVW-1*, *ERVH48-1* and its binding partner, in cells and tissues representing early human pregnancy. Finally, we have performed proof of principle studies to determine whether suppressyn can inhibit fusion between trophoblast and endometrial epithelial cell types in vitro. 

## 2. Methods

### 2.1. Decidual Tissue Samples

Non-pregnant decidual tissues were obtained from premenopausal woman who underwent hysterectomy during their follicular phases for symptomatic uterine fibroids (*n* = 3). Decidual tissues from early pregnancy were obtained from a patient who underwent surgical therapy for cervical carcinoma at 7 weeks of gestation with *in toto* removal of uterus and cervix with pregnancy tissues in situ as described previously [16]. Enrolled subjects were otherwise healthy, and the pregnant subject had an otherwise uncomplicated pregnancy. All collections were approved by the Ethical Committee for Human Genome Research of the Hiroshima University (hi-222) and appropriate informed consent was obtained from all participants. All research was performed in accordance with relevant guidelines/regulations. 

### 2.2. Cell Lines

HTR-8/SVneo cells (referred to as HTR8 cells throughout for simplicity) [19,20] were a kind gift from Professor Charles Graham of the Department of Anatomy and Cell Biology at Queen’s University, Kingston, ON, Canada. The endometrial Ishikawa cell line was generously supplied by Dr Susan Nagel at The University of Missouri. HTR8 and Ishikawa cells were cultured in DMEM (041-29775: Fuji film, Tokyo, Japan) supplemented with 10% FBS (Fetal Bovine Serum:Gibco-Thermo Fisher scientific, Waltham, MA, USA) at 37 °C in humidified 5% CO_2_/20% O_2_. Methods for the establishment of HTR8 cells stably expressing SUPYN (HTR8-SUPYN) and the matched vector-only control (HTR8-Vec) and their culture in the same medium with the addition of 1 μg/mL puromycin have been described previously [15]. 

### 2.3. Immunohistochemistry

Tissue blocks were prepared from formaldehyde-treated tissue using standard methods. Slides with 4 μm sections were heated at 60 °C for 15 min and then standard deparaffinization with xylene was performed. Antigen retrieval was carried out using an electric kettle at 98 °C for 40 min in 0.2 M citrate buffer at pH 6.0. Inactivation of endogenous peroxidase activity was performed in 0.3% H_2_O_2_/methanol at room temperature for 20 min. Sections were blocked with donkey serum for rabbit antibodies at room temperature for 30 min. The anti-ASCT2 (SLC1A5) antibody (#8057: CST Danvers, Danvers, MA, USA) was diluted 1/2000 0.1% BSA/PBS and Rabbit isotype control (#3900: CST Danvers, Danvers, MA, USA) was used at the same concentration. Sections were incubated with primary antibody overnight at 4 °C. Slides were washed three times in PBS and exposed to biotinylated anti-rabbit antibody (VECTORSTAIN Elite ABC HRP kit: PK-6101:VECTOR Laboratories, Burlingame, CA, USA) at room temperature for 30 min, then to the streptavidin-HRP at room temperature for an additional 30 min (Streptavidin-Biotin Complex Peroxidase kit: 30462-30: nakalai tesque, Kyoto, Japan). Color development utilized DAB (Peroxidase Stain DAB kit: 25985-50: nakalai tesque, Kyoto, Japan) and hematoxylin counterstaining was carried out in the standard fashion at room temperature for 5 min.

### 2.4. Data Availability

The data presented in Figure 2A are available at http://data.teichlab.org (maternal–fetal interface), accessed on 14 November 2018 [21] and BioProject ID PRJNA492324 [22], respectively. The data presented in Figure 2B are available in Gene Expression Omnibus (GEO) under accession numbers GSE136447 [23].

### 2.5. Cell Staining with CellVue Claret and CSFE, Transient Transfection of syn1 and Co-Culture

HTR8 cells (HTR8, HTR8-Vec and HTR8-SUPYN) were trypsinized and washed twice in PBS. For PKH26 (red) staining, the cell suspension (3 × 10^6^ cells) was exposed to 600 μL of Dilution C reagent with 7 × 10^−7^ M CellVue Claret (MINCLARET and MIDCLARET: Sigma: Saint Louis, MO, USA). After 5 min incubation, excess PKH26 dye was absorbed with 600 μL of FCS and cells were washed 3 times with 10 mL of conditioned medium. 1.5 × 10^5^ cells/well were then seeded in 12 well plates and incubated at 37 °C in 5% CO_2_/20% O_2_ condition. After 24 h of incubation, cells were exposed to 100 ng of a plasmid vector driving expression of syn1 for transient transfection using Lipofectamine 2000 (11668027: Thermo Fisher Scientific, Waltham, MA, USA) and incubated for 6 h as described previously in detail [15]. Cells were washed 3 times with 2 mL conditioned medium and then placed into co-culture with 7 × 10^4^ cell CSFE (green)-stained Ishikawa cell. For CSFE staining, 3 × 10^6^ Ishikawa cells were incubated with 0.7 μg/mL CSFE solution (341-07401: Dojindo, Kumamoto, Japan) for 30 min at 37 °C in PBS. After absorption with an equal volume of FCS, CSFE-exposed Ishikawa cells were washed 2 times with 30 mL of conditioned medium. Following 24 h incubation, co-cultured cells were fixed with 2% paraformaldehyde and nuclei were stained with 0.5 μg/mL of Hoechst 33342 (H342 Dojindo, Kumamoto, Japan). Fluorescent signals were detected using a fluorescence microscope (BZ-X710; KEYENCE Japan, Osaka, Japan). For clarity, these methods have been summarized in visual format in Appendix A.

## 3. Results

We have previously demonstrated that EVT express SUPYN protein [15,16]. Here, we have analyzed the localization of the syn1- and SUPYN-binding partner SLC1A5 in the non-pregnant endometrium and decidua of early pregnancy (Figure 1). Immunohistochemical staining of 3 non-pregnant endometrial samples representing the mid-proliferative phase of the menstrual cycle demonstrated strong expression of SLC1A5 in the glandular epithelium (Figure 1A), but little to no consistent detectable expression in endometrial stromal cells. In contrast, SLC1A5 protein expression in the decidualized endometrium of our unique in situ sample of a 7-week pregnancy was quite pronounced in endometrial stromal cells (Figure 1B) but weak in the endometrial glandular epithelium. These results confirmed prior work by our laboratory [16] and together place the ERV-derived profusogenic and anti-fusogenic placental proteins, syn1 and SUPYN, and their common receptor in close proximity within the decidua of early pregnancy.

We next analyzed several publicly available single cell RNA sequencing (scRNAseq) databases representing early pregnancy tissue samples and in vitro decidualization models to confirm our immunohistochemical results (Figure 2A, upper panel). As seen in the Vento-Tormo dataset [21], derived from first trimester terminations of pregnancy, *SLC1A5* is fairly widely transcribed in tissues from the maternal-fetal interface. This certainly includes both subpopulations of EVT (EVT > EVTp) and all three subsets of decidual stromal cells (DS1, DS2 and DS3) at fairly equal levels. Consistent with our IHC data from the sample from 7 weeks of gestation, transcription in the epithelial glandular cells in their dataset was very low to absent. Interestingly, *SLC1A5* was noted to be transcribed in maternal endothelial cells (Endo m) which likely come from decidual venules, veins, capillaries, arterioles, and arteries, including the maternal spiral arteries, all of which, along with maternal decidual glands, can be invaded, and presumably remodeled, by EVT. Additionally, consistent with our IHC results, *ERVW-1* and *ERVH48-1* transcription was largely limited to trophoblast subpopulations, including STB, CTB and, relevant to this study, EVT. The scRNAseq dataset from Suryawanshi et al. [22] (Figure 2A, lower panel) also represents first trimester pregnancy tissues from terminations of pregnancy. Like the data from Vento-Tormo, that of Suryawanshi showed *ERVW-1*, *ERVH48-1* and *SLC1A5* transcription in all TB subsets, although relative levels of transcription varied among these subtypes. As was also reported by Vento-Tormo et al., *SLC1A5* was transcribed by a wide variety of decidual cell types with decidual stromal cell transcription markedly greater than that in the glandular epithelium.

We then examined the two publicly available scRNAseq datasets to determine transcriptional levels of *ERVW-1* and *ERVH48-1* in the trophectoderm and early TB derivatives of human blastocysts. One of these manuscripts [24] specifically included HERV env expression evaluation in their own analyses and will therefore be discussed in the discussion section below. We assessed these transcriptional levels to determine whether the first cells to interact with the maternal endometrial glandular epithelium at blastocyst apposition attachment and both the epithelium and decidualized stroma upon further implantation might display the ERV envelope-derived partners of SLC1A5 known to be expressed at the RNA and protein levels in the decidua as demonstrated above. As shown in Figure 2B, re-analysis of the dataset from Xiang et al., showed high levels of *ERVW-1* transcription in preSTB and STB subpopulations, but lower levels in early EVT and EVT. They found the converse pattern of expression for *ERVH48-1*. 

As proof of principle that heterotypic cell–cell fusion can occur and can be blocked in cell types relevant to the maternal and fetal cells in the human decidua, we designed an in vitro model for trophoblast/endometrial cell ERV-mediated fusion and its ERV-mediated inhibition. We used HTR8 cells to model EVT because they were derived from primary EVT cells [19,25] and are known to lack expression of syn1 and SUPYN [15] and endometrial adenocarcinoma-derived Ishikawa cells to represent endometrial glandular epithelial cells [26,27]. Ishikawa cells do not endogenously express syn1 but are known to express SLC1A5 [15]. HTR8 cells with and without stably transfected *ERVH48-1* were transiently transfected with a vector driving *ERVW-1* to drive homotypic and heterotypic cell fusion in HTR8/Ishikawa cell co-cultures. HTR8-derived cells were stained red with PKH26, and Ishikawa cells stained green with CSFE to allow visualization of heterotypic cell fusion. Control experiments using heterotypic co-cultures of cells lacking syn1, including Ishikawa cells and HTR8 parental cells (HTR8), HTR8 cells stably transfected with *ERVH48-1* control vector alone (HTR8-Vec) and HTR8 cells stably transfected with a vector driving expression of *ERVH48-1* (HTR8-SUPYN) are shown in Appendix A. As expected, in the absence of *ERVW-1* transcription/syn1 expression, whether in mock-transfected or HTR8-derived cells transfected with the control vector for *ERVW-1* only (pFlag-control), neither heterotypic nor homotypic cell–cell fusion was seen. Red cells and green cells, while sometimes clustered, are distinct and are not fused with themselves and their heterotypic counterparts. When *ERVW-1* (pFlag-syncytin-1) was transiently expressed in HTR8 parental and HTR8-Vec cells lacking *ERVH48-1* transcription, homotypic and heterotypic cell fusion was demonstrated by the presence of multinucleated cells. Multinucleated red cells (HTR8-HTR8 fusion) and orange cells (HTR8-Ishikawa fusion) were both detected as predicted. We also detected homotypic Ishikawa-Ishikawa cell fusion (multinucleated green cells) which was unexpected and thought to represent carry-over of the transiently transfected *ERVW-1* expressing plasmid during co-culture (Figure 3). In contrast, when *ERVW-1* was transiently expressed in HTR8-SUPYN cells that stably express SUPYN protein and these cells were co-cultured with Ishikawa cells, we detected neither red nor orange-colored multinucleated cells, demonstrating that HTR8 homotypic and HTR8-Ishikawa fusion were abrogated (Figure 3). Interestingly, while most cells in cultures in which HTR8-SUPYN cells were present remined unfused, we did detect several green-colored multinucleated cells, indicating that homotypic Ishikawa cell fusion was still occurring.

## 4. Discussion

SUPYN protein expression is specific to human placenta and has been previously shown to inhibit syn1-induced homotypic cell fusion between the trophoblast cell types present in the villous placenta, likely regulating cytotrophoblast fusion into STB [15,16]. We have previously reported that placental SUPYN protein can be detected in unfused trophoblast cells, including villous cytotrophoblast (CTB) and cell column cytotrophoblast (CCC) in vivo, but also in EVT [15,16]. In hypothesizing a role for SUPYN in the extravillous decidua, we speculated that SUPYN might negatively regulate not only homotypic EVT fusion, but also heterotypic fusion of EVT with decidual glandular and stromal cells and possibly even maternal endothelial cells. In this study, we began by demonstrating that cells expressing the relevant pro- and anti-fusogenic proteins, syn1 and SUPYN, respectively, and their common receptor, SLC1A5 are temporally and geographically in close proximity within the maternal decidua during the first trimester of pregnancy in vivo. Knowing from prior studies [6,11,14,28] that syn1-mediated homotypic and heterotypic cell–cell fusion required syn1 expression by one cell and SLC1A5 by the other, we first needed to assess expression of these binding partners on cells in the decidua that could come into direct contact. Based on our data showing that SUPYN could have effects via its intracellular and secreted forms [15,16], we hypothesized that the SUPYN-expressing cell may have inhibitory effect on both proximate and possibly nearby cells. 

To demonstrate expression of relevant receptor-binding partners in the decidua, we combined IHC results from a sample from the first trimester of pregnancy with data from publicly available scRNAseq analyses of first trimester tissues from the human maternal-fetal interface. Expression of SLC1A5 in the decidua has been previously reported using IHC [18]. We also detected its expression in our sample from week 7 of pregnancy, noting that expression was stronger in the decidual stromal cells than in the glandular epithelium. 

Interestingly, there was a converse pattern in endometrial samples from non-pregnant women, with strong expression in the glandular epithelium and weaker expression in the endometrial stroma. Single-cell RNAseq analyses of first trimester human pregnancy tissues verified our IHC data on SLC1A5, showing expression in maternal decidual cells, with stromal expression greater than epithelial expression. These same scRNAseq analyses show that *ERVW-1* and *ERVH48-1* are both expressed in EVT, confirming IHC from ourselves and others [29,30]. Based on these results, we hypothesized that EVT cells that invade the maternal decidua might fuse with other EVT and with maternal decidual cells and that the presence of SUPYN at these sites may suppress such fusion. We further speculate that a decrease in glandular epithelial cell expression of SLC1A5 in pregnant women may reflect an escape mechanism for untoward EVT/glandular epithelial cell fusion. 

Results from publicly available datasets obtained through analyses of human extended blastocyst cultures further supported IHC and scRNAseq data on the expression of *ERVW-1*, *SLC1A5* and *ERVH48-1* in tissues obtained from mid-first trimester pregnancies. West et al. [24] reported on the expression of HERV-derived genes in their scRNAseq dataset examining human blastocysts grown for up to 12 days in culture. They show that *ERVW-1* transcription decreases from 10 dpf–12 dpf in STB precursors and STB but increases from 8 dpf–12 dpf in migratory EVT precursors. In contrast, these same investigators reported strong and increasing expression of *ERVH48-1* in CTB from 8 dpf–12 dpf and increasing expression of *ERVH48-1* in EVT precursors (labeled MTB and pre-MTB in their manuscript). Although the human extended blastocyst culture dataset from Xiang et al., analyzed in Figure 2B reveal expression patterns that differ across time, both show that human trophectoderm and primitive trophoblast derivatives have patterns of expression that localize ERV envelope proteins and their cognate receptor to cell types that are in close proximity at the time of implantation.

Interestingly, data from scRNAseq analyses of first trimester pregnancy tissues also localized transcription of *SLC1A5*, the SUPYN/syn1 receptor, to the maternal endothelial cells, suggesting the exciting possibility that SUPYN may regulate heterotypic fusion between syn1-expressing EVT and SLC1A5-expressing maternal spiral arteries, arterioles, veins, venules and glands during their invasion and remodeling of these structures in early pregnancy. The risk for unwanted EVT/endothelial fusion may, in fact, be particularly elevated at the sites of maternal spiral arteries and arterioles, in which the oxygen content is known to be higher than in the surrounding decidual tissues [31,32]. We previously reported the presence of fused EVT cells within a maternal arteriole in the same 7-week placental sample used for the experiments reported in this manuscript [16]. We have also shown in an in vitro model of term trophoblast syncytialization that syn1 transcription increases and SUPYN expression decreases in a dose-dependent manner as surrounding O_2_ concentrations increase [16]. This suggests that syn1 may increase in syn1-expressing cells as they near the maternal decidual arteries and arterioles. While these changes may be more relevant to placental health and disease in the first trimester during placental development than at term, these findings may still accurately reflect those occurring earlier. To this point, ambient oxygen concentrations had little effect on transcription of SLC1A5 in this model and had a converse relationship with *ERVH48-1* transcription. Effects on protein expression were not assessed in these experiments and may differ from transcriptional changes.

As a proof-of-principle exercise, we next assessed the potential for SUPYN to inhibit syn1-induced homotypic and heterotypic fusion between trophoblast and maternal decidual cells at the maternal-fetal interface. To mimic a human EVT cell, we transiently transfected syn1 into HTR8 cells, which themselves were derived from a primary EVT cell [19,33] but lack endogenous syn1 protein expression [15]. To assess the effects of SUPYN on syn1-induced cell–cell fusion, we transiently expressed *ERVW-1* in three HTR8 cell subtypes, one that was mock-transfected (HTR8), one stably transfected with a control vector (HTR8-Vec) and one that was stably transfected with the same vector driving the transcription of *ERVH48-1*(HTR8-SUPYN). Transiently transfected syn1-expressing HTR8 cells with and without stably expressed *ERVH48-1* were then co-cultured with a heterotypic human endometrial adenocarcinoma cell line, Ishikawa cells, which endogenously express the syn1- and SUPYN-binding partner, SLC1A5, but do not endogenously express syn1 or SUPYN [15]. Cell–cell fusion under co-culture conditions was then assessed in two ways: (1) phase contrast microscopy was used to assess intercellular boundaries to allow for detection of multinuclear cells regardless of cell color and (2) red (HTR8) and green (Ishikawa) dyes, when combined with phase contrast images, allowed for determination of homotypic versus heterotypic fusion. As hypothesized, transiently transfected syn1 on HTR8 cells drives fusion between all types of SLC1A5-expressing cells, including homotypic HTR8/HTR8 fusion (red multinucleated cells) and heterotypic HTR8/Ishikawa fusion (orange multinucleated cells). We were surprised, however, that homotypic fusion was also seen between Ishikawa cells (green multinucleated cells), as they do not endogenously express syn1. We believe this is most likely the result of inadvertent carryover of the transient transfection plasmid to some Ishikawa cells despite washing of transfected HTR8 subtypes prior to introduction into co-culture. This result did, however, provide additional insight into the mechanism of SUPYN-mediated fusion inhibition in our co-culture model (see below). When HTR8-SUPYN cells were partnered with Ishikawa cells in co-culture, we noted an inhibition of homotypic cell–cell fusion between HTR8-SUPYN cells and inhibition of heterotypic cell–cell fusion between HTR8-SUPYN and Ishikawa cells. Interestingly, the presence of HTR8-SUPYN cells in co-culture did not completely abrogate homotypic fusion between Ishikawa cells.

These proof of principle experiments suggest that several SLC1A5-expressing maternal cells within the decidua have the potential to fuse with syn1-expressing EVT and that EVT have the potential to fuse with themselves during decidual invasion. While the origin of the trophoblast giant cells seen in the basal decidua in humans remains incompletely defined [34,35,36], they may represent fairly rare occurrences of EVT—EVT fusion [37]. That said, despite close proximity and receptor ligand pairs that could allow for such fusion, most EVT in the decidua fuse neither with other invasive EVT nor with cells of the maternal endometrium during pregnancy, including glandular epithelial cells, decidual stromal cells, endovascular cells and decidual immune cells. We speculate that SUPYN may have a role in inhibition of fusion in at least some of these sites. Further, dysfunction in such inhibition may be linked to dysfunction in placental form and or function. It is certainly well-documented that altered regulation of syncytialization in the villous placenta may be integral to the increased STB turnover and release in conditions such as pre-eclampsia and fetal growth restriction and that syn1 may be involved in such dysfunction [38,39,40,41,42].

From a more mechanistic standpoint, we have previously described two ways in which SUPYN inhibits fusion events mediated by syn1-SLC1A5 interactions (Figure 4). In the first [15], SUPYN secreted (it has no mechanism for cell surface expression) by EVT in the decidua directly blocks or alters ligand binding and/or post-binding signaling between cell surface-expressed syn1 on EVT and SLC1A5 on another EVT cell or a maternal cell in the decidua. Such receptor inhibition is well-described for several retroviral envelope proteins and, indeed, SLC1A5, also known as RD114 in this area of study [43], is a prime example of a cell surface receptor that facilitates entry of a family of viruses that exhibit such receptor inhibition [44]. In the second mechanism [16], intracellular SUPYN interacts with SLC1A5 during its processing for cell surface expression and alters it glycosylation, thereby affecting its cell surface stability and/or ligand-binding capacity.

Both mechanisms may be at play in our co-culture experiments, particularly when homotypic HTR8 fusion and heterotypic HTR8-Ishikawa fusion is inhibited, as cell partners in these interactions include the HTR8 cell that simultaneously synthesizes SUPYN and SLC1A5 and could thereby alter intracellular SLC1A5 properties but can also secrete SUPYN locally to alter HTR8-expressed cell surface syn1 and SLC1A5 expressed on the Ishikawa fusion partner. We speculate that the lack of complete abrogation of homotypic Ishikawa fusion results from the necessity for fusion inhibition to rely solely on secreted SUPYN. We previously showed a 30% decrease in cell fusion rate using high concentrations of exogenously added purified secreted SUPYN protein [15]. This relatively weak effect despite high concentrations, combined with likely lower concentration of locally secreted SUPYN when the secreting HTR8 cell is not directly involved in the cell fusion event, would have diminished our ability to detect changes in fusion inhibition. We hypothesize that the primary mechanism by which SUPYN inhibits cell–cell fusion involves it direct interactions with its receptor in the cell cytoplasm, and that secretory SUPYN is a less important and secondary mechanism that is limited to nearby neighboring homotypic or heterotypic SLC1A5-expressing cells.

Despite the importance of an improved understanding of factors involved in cell–cell interactions and fusion events during very early human placentation, including blastocyst implantation and invasion into the maternal decidua and spiral arteries, studies have been limited and this part of human pregnancy remains largely a “black box”. Ethical and logistical considerations have combined to nearly obstruct access to human placental and decidual tissues from the preclinical, peri-implantation stage of pregnancy. Here, we had the opportunity to study an in situ first-trimester pregnancy specimen to detect SLC1A5 protein expression using IHC. This too does not directly address peri-implantation events, but it represents a rare look into pregnancy soon after that period. Similarly, although statistically significant sample numbers were not available, publicly available scRNAseq data analyses of early first trimester pregnancy samples supported our IHC results and expanded upon them to address TB subtype and decidual cell subtype-specific expression of fusogen, anti-fusogen and receptor ligand expression patterns. Patterns of decidual expression in the glands versus the stroma also support a biological function at the time of implantation. We similarly examined RNAseq databases from in vitro blastocyst culture to localize ERVW-1 and ERVE48-1 transcription. While we have shown expression of this fusogen and anti-fusogen and their shared receptor in cells and tissues temporally bracketing the black-box period, formal proof of expression at the time of implantation remains elusive.

Our in vitro co-culture model of human EVT cells (HTR8 cells) and human endothelial cells (Ishikawa cells) revealed a central role for syn1 and SUPYN in heterologous syn1- driven cell–cell fusion at the maternal-fetal interface. Although this is a proof-of-principle demonstration, the cells used are derived from a cancer (Ishikawa) or are transformed (HTR8) and therefore cannot precisely model primary cell interactions at the time of implantation. Technically, we were limited by the colors available to use during heterologous cell fusion experiments so we could not assess cell boundary markers (such as e-cadherin or zona occludins-1) expression as well as red and green cell dyes and nuclear staining. We must therefore show heterologous fusion through the mixture of red and green dyes. We have included as Appendix A, a depiction of Ishikawa monocultures driven to fusion by two different concentration of ERVW-1 transient transfection and stained cell boundaries with anti-e-cadherin antibody to demonstrate syncytialization.

Our experiments also cannot formally exclude that Ishikawa cells are fusing with themselves through interactions other than those mediated by carry-over syn1. Appendix A suggests that syn1 is driving this fusion. Further, Ishikawa cells, while known to express the syn2 receptor MFSD2, do not express syn2 and these cells do not commonly fuse during routine culture.

In summary, while the importance of balanced cell–cell fusion is well-described for the formation of STB in the villous placenta, there is potential for the same processes to occur between trophoblast and non-trophoblast cell types in the maternal decidua. We have previously proposed that SUPYN may be important for the control of fusion in the villous placenta and here suggest that it may also have a role at extravillous sites within the maternal-fetal interface. These could include inhibition of fusion off the primitive trophoblast with glandular epithelium at the time of implantation, of EVT with decidual endometrial cells (glandular and stromal) during invasion and of EVT/endovascular TB with the endothelium of a variety of vascular structures within the endometrium, the best described of which occurs during the remodeling of the maternal spiral arteries. This latter process, when altered, is closely associated with poor pregnancy outcomes. If SUPYN plays a central role in any of these processes, a better understanding of its function could promote insights into predictive markers and novel approaches to the diagnosis and treatment of diseases of abnormal placentation.

## Figures and Tables

**Figure 1 ijms-22-10259-f001:**
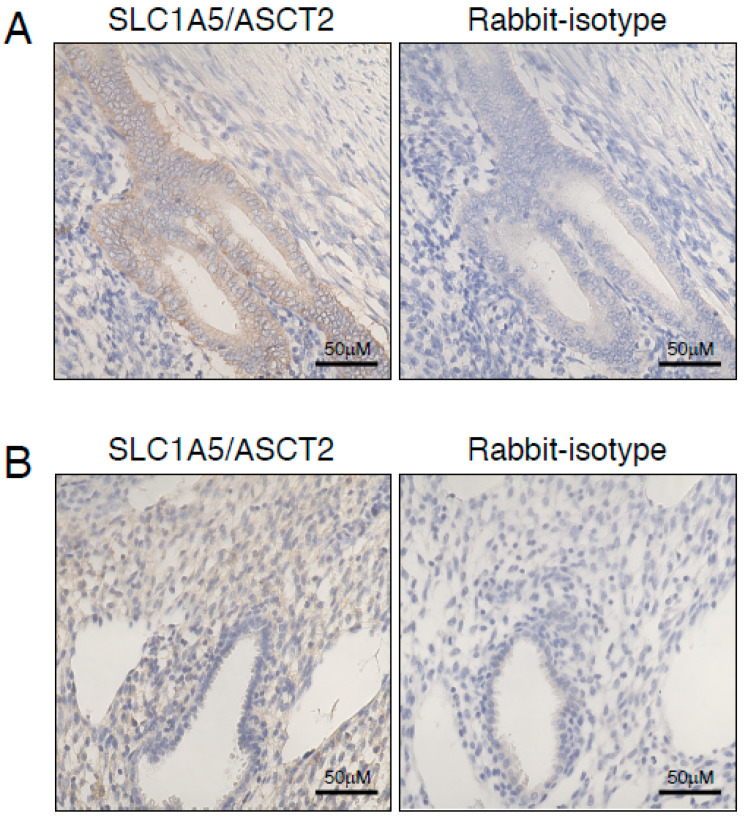
Expression of SLC1A5/ASCT2 in endometrial and decidual tissues. Immunohistochemical staining was performed for SLC1A5 using endometrial tissues from non-pregnant women in the follicular phase (**A**) and a sample of a placenta with decidual tissues obtained from a 7-week pregnancy (**B**). Scale bars are as indicated.

**Figure 2 ijms-22-10259-f002:**
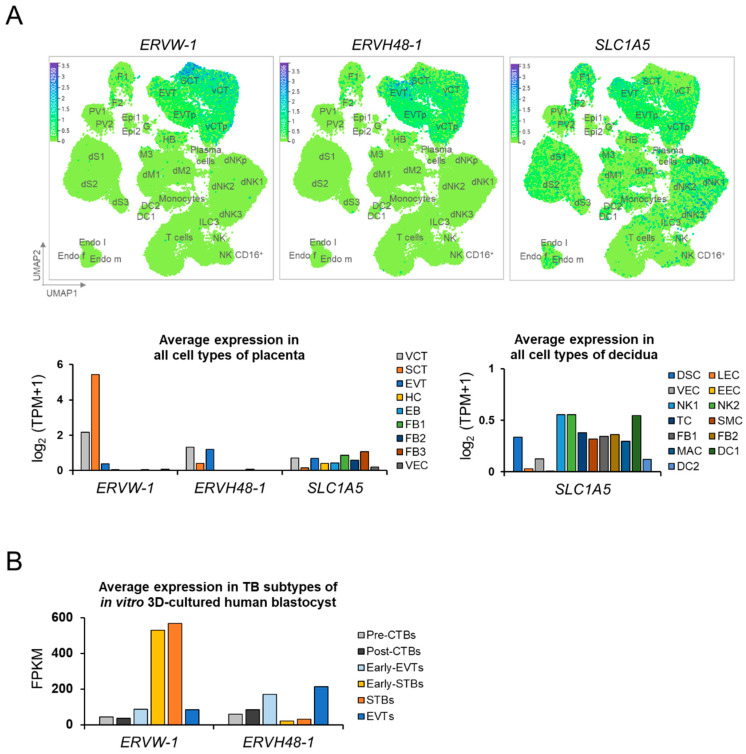
In silico analysis of Syncytin-1 (*ERVW-1*), SUPYN (*ERVH48-1*), and/or *SLC1A5*/ASCT2 expression in the human first-trimester placenta/decidua and in the TB subtypes of extended 3D human blastocyst cultures. (**A**, upper) UMAP visualization of the log-transformed, normalized expression levels in placental and decidual cell clusters of the first trimester (6 to 14 weeks of gestation; placenta, *n* = 5; decidua, *n* = 11; blood, *n* = 6). Data are taken from a publicly available datasets found online at http://data.teichlab.org (maternal-fetal interface) provided by Vento-Tormo et al. (accessed on 14 November 2018) [21]. A graph-based clustering analysis was applied for distinct cell clusters and cluster-specific marker genes were used to annotate the clusters. DC, dendritic cells; dM, decidual macrophages; dS, decidual stromal cells; Endo, endothelial cells; Epi, epithelial glandular cells; F, fibroblasts; HB, Hofbauer cells; PV, perivascular cells; SCT, syncytiotrophoblast; VCT, villous cytotrophoblast; EVT, extravillous trophoblast; f, fetal; ILC, innate lymphocyte cells; l, lymphatic; m, maternal; p, proliferative; M3, maternal macrophages; G, granulocytes. (**A**, lower) Average transcript expression in all cell types derived from placenta and decidua of the first trimester (6 to 11 weeks of gestation; villi, *n* = 8; decidua, *n* = 6). scRNAseq data are from a publicly available dataset provided by Suryawanshi et al. [22]. A graph-based clustering analysis was applied for distinct cell clusters. VCT, villous cytotrophoblasts; SCT, syncytiotrophoblasts; EVT, Extravillous trophoblasts; HC, Hofbauer cells; EB, erythroblasts; FB, fibroblasts; VEC, vascular endothelial cells; DSC, decidualized stromal cells; LEC, lymphatic endothelial cells; VEC, vascular endothelial cells; EEC, endometrial epithelial cells; NK, natural killer cells; TC, T cells; SMC, smooth muscle cells; FB, fibroblasts; MAC, maternal macrophages; DC, dendritic cells; TPM, Transcripts Per kilobase Million. (**B**) Average transcript expression in the trophoblast (TB) subpopulations of in vitro three-dimensional (3D) cultured human blastocysts from pre-implantation stage to 14 days post-fertilization (dpf) (total embryos, *n* = 42; 6 dpf, *n* = 8; 7 dpf, *n* = 5; 8 dpf, *n* = 6; 9 dpf, *n* = 6; 10 dpf, *n* = 6; 12 dpf, *n* = 5; 14 dpf, *n* = 6). scRNAseq data are from a publicly available dataset provided by Xiang et al. [23]. Cell types were defined based on classical lineage-specific marker expression and developmental time. CTB, cytotrophoblasts; EVT, Extravillous trophoblasts; STB, syncytiotrophoblasts; FPKM, Fragments Per Kilobase of transcript per Million mapped reads.

**Figure 3 ijms-22-10259-f003:**
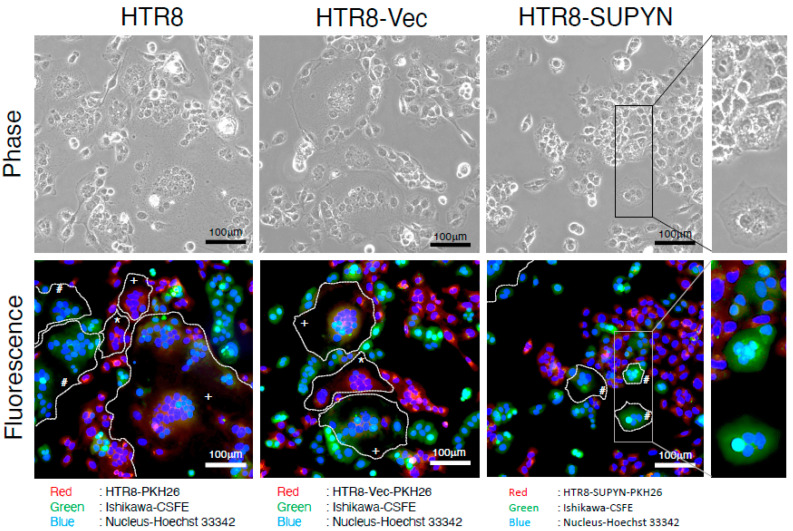
In vitro analysis of syn1-induced homotypic and heterotypic trophoblast/endometrial epithelial cell fusion and the effects of SUPYN. HTR8 trophoblast cells, HTR8 cells stably transfected with a control vector (HTR8-Vec) and HTR8 cells stably transfected with a vector driving SUPYN expression (HTR8-SUPYN) were stained with a red dye and Ishikawa endometrial cells with a green dye and placed in co-culture. All cell nuclei were counterstained blue with Hoechst 33342. In the left and middle panels, HTR8 (red; far left) and HTR8-Vec (red, middle left) TB cells were transiently transfected with a plasmid driving the expression of syn1 prior to co-culture with Ishikawa endometrial cells (green). In the right panels, HTR8-SUPYN cells were transiently transfected with a plasmid driving the expression of syn1 prior to co-culture with Ishikawa endometrial cells (green). Phase contrast images are presented for comparison to aid in distinguishing cell boundaries. A magnified area is shown in the far-right column. Red-red HTR8 homotypic fusion is depicted with asterisks (*); green-green Ishikawa homotypic fusion is depicted with a pound sign (#) and orange heterotypic HTR8/Ishikawa cell fusion with a plus sign (+).

**Figure 4 ijms-22-10259-f004:**
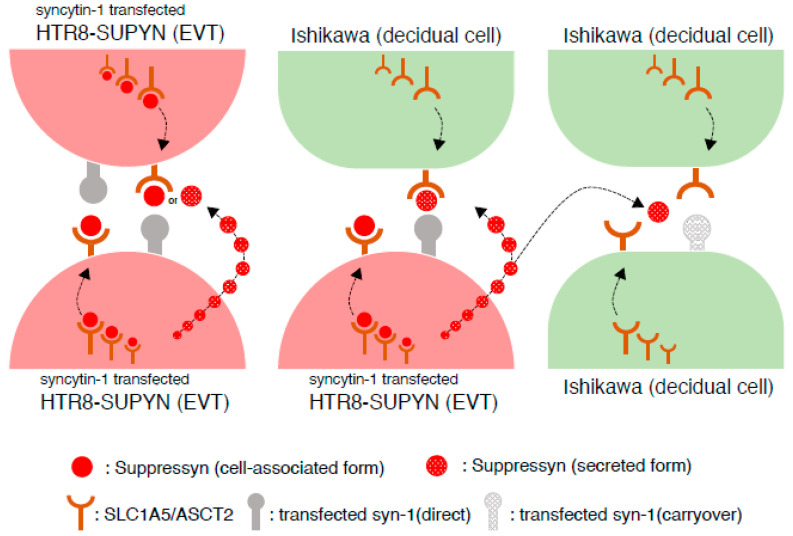
Mechanism of SUPYN interaction with SLC1A5/ASCT2 in allogenic and heterogenic cells. Three types of the many possible decidual cell interactions in the human decidua that are modeled by our in vitro experiments are illustrated: EVT-EVT (left), EVT-decidual epithelia or decidual stroma (decidua) and (middle) and decidual-decidual (right). In EVT-EVT homotypic fusion seen as red-colored, multinuclear HTR8 homotypic fusion (Figure 3), it is likely that SUPYN forms a heterotypic protein complex with SLC1A5 in the HTR8 cell cytoplasm and suppresses normal receptor-binding with syn1 expressed on the cell surface of a partner HTR8 cell. In EVT-decidual epithelial cell heterotypic fusion, seen as orange-colored multinuclear HTR8-Ishikawa heterotypic cell fusion in Figure 3, green Ishikawa cells do not endogenously express SUPYN, and intracellular processing of SLC1A5 will not be affected by such interactions. However, secretory SUPYN protein is released from partner HTR8-SUPYN cells; binds to Ishikawa cell-expressed SLC1A5 and inhibit the profusogenic actions of syn1. In decidual epithelial cell homotypic fusion seen as green colored multinuclear Ishikawa homotypic fusion in Figure 3, green Ishikawa cells that express SLC1A5 should express neither syn1 nor SUPYN and therefore would not be predicted to undergo homotypic fusion. In our experiments, we hypothesize that such fusion was driven by inadvertent carryover of syn1-expressing plasmid during co-culture with transiently transfected HTR8 cells. We further suggest that this aberrant homotypic Ishikawa cell fusion was not blocked by SUPYN secreted from HTR8-SUPYN cells in co-culture due to an insufficient local SUPYN concentration at the site if Ishikawa-Ishikawa cell interactions. Symbols for cell-associated and secreted forms of SUPYN, transfected syn1 (direct and carryover) and SLC1A5 depicted in the diagram are identified at the bottom of the figure.

## Data Availability

Publicly available datasets were analyzed in Figure 2 of this study. The data presented in Figure 2A are available at http://data.teichlab.org (maternal–fetal interface), accessed on 14 November 2018 [21] and BioProject ID PRJNA492324 [22], respectively. The data presented in Figure 2B are available in Gene Expression Omnibus (GEO) under accession numbers GSE136447 [23].

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
