# Peer review of "Could the Human Endogenous Retrovirus-Derived Syncytialization Inhibitor, Suppressyn, Limit Heterotypic Cell Fusion Events in the Decidua?"

_ijms, 2021, doi:10.3390/ijms221910259_

Round 1
Reviewer 1 Report
The manuscript by Sugimoto et al concerns an analysis of the effect of suppressyn/SUPYN expression on cellular fusion events during human placentation. First, the authors quantified transcripts for syncytin, suppressyn and their receptor, ASCT2/SLC1A5, expression in diverse cell types involved in early pregnancy, from various databases. And although transcript levels of course are not a direct measure of protein production, the results are in line with what would be expected based on earlier work. Then, the authors proceed to perform cell-culture experiments, where they attempt to show the effect of suppressyn expression of fusion between two cell types that resemble the extravillous trophoblasts (HTR-8 cell line) and endometrial glandular epithelial cells (Ishikawa cell line). However, here the results are less clear, and additional information is needed:
- How often did you repeat the cocultures shown in Fig. 3? Can the fusion level of the cocultures be quantified? So that it will be more easy to show eventual blocking by suppressyn? Now all we have is a comparison by eye between only two panels, and the orange colour is difficult to discern.
- Is the vector-expressed suppressyn protein retained intracellularly or secreted? Did you check for suppressyn expression in the transfected cells?
- In what ratio were HTR-8 and Ishikawa cells cocultured? From Fig. 3 and the supplementary figure it appears that there are much less green cells than red cells. What was the rationale behind the chosen ratio? Did you also assess other ratios?
- About the unexpected fusion of Ishikawa cells: when only Ishikawa cells are cultured, is there any fusion-like process visible? In the supplementary figure, green cells are found mostly together, but all panels concern cocultures. Could the unexpected fusion of Ishikawa cells (Fig. 3) possibly be due to syn2 & MFSD2 expression? Or some other interaction? When fusion occurs due to plasmid carryover, there should be different levels of fusion between experiments. Did you try DNAse treatment before coculture to prevent contamination?
Minor comments:
Please decide on which name to use for ASCT2/SLC1A5, as in for instance in the legends of fig. 2 SLC1A5 is used, but the results section (pages 5) continues with the use of ASCT2, then again SLC1A5 (page 6), etc.
Line 63: please write out EVT at first use
Fig. 2: what exactly are (TPM+1) expression levels in lower panel A?
Lines 151, 153: some unit is missing in ‘600 l’
Line 159-160: twice ‘cell’ in coculture description
Line 331: ‘that at term’ is likely ‘than at term?
Reviewer 2 Report
The manuscript focuses on assessing the significance of human endogenous retroviruses for placental development. The manuscript is well written, interesting and susceptible for publication.
Unfortunately, there are some limitations:
Lines 66-78 are connected to the study results. A majority of his part should be located in Discussion. Please concentrate only on aims f this manuscript here.
Please provide the simplified graph of the study stages (i.e. the following procedures).
Please indicate the limitations of your study in Discussion. Please include ethical considerations / the examined decidual tissues of a relatively small number of patients / inability to assess decidual tissues from healthy women with physiological pregnancies.
Round 2
Reviewer 1 Report
All my comments have been addressed in the revised version of the manuscript by Sugimoto et al. The paper has improved significantly now that additional explanations and supplementary figure 3 have been added, because it is not convenient to have to search for basic information in the supplementary data of an article published years ago.